# Circularly polarized electroluminescence from a single-crystal organic microcavity light-emitting diode based on photonic spin-orbit interactions

Jichao Jia[1,5], Xue Cao[1,5], Xuekai Ma[2], Jianbo De[3], Jiannian Yao[3], Stefan Schumacher[2,4], Qing Liao[1] ✉ & Hongbing Fu[1] ✉

Circularly polarized (CP) electroluminescence from organic light-emitting diodes (OLEDs) has aroused considerable attention for their potential in future display and photonic technologies. The development of CP-OLEDs relies largely on chiral-emitters, which not only remain rare owing to difficulties in design and synthesis but also limit the performance of electroluminescence. When the polarization (pseudospin) degrees of freedom of a photon interact with its orbital angular momentum, photonic spin-orbit interaction (SOI) emerges such as Rashba-Dresselhaus (RD) effect. Here, we demonstrate a chiral-emitter-free microcavity CP-OLED with a high dissymmetry factor ($g_{EL}$) and high luminance by embedding a thin two-dimensional organic single crystal (2D-OSC) between two silver layers which serve as two metallic mirrors forming a microcavity and meanwhile also as two electrodes in an OLED architecture. In the presence of the RD effect, the SOIs in the birefringent 2D-OSC microcavity result in a controllable spin-splitting with CP dispersions. Thanks to the high emission efficiency and high carrier mobility of the OSC, chiral-emitter-free CP-OLEDs have been demonstrated exhibiting a high $g_{EL}$ of 1.1 and a maximum luminance of about 60000 cd/m², which places our device among the best performing CP-OLEDs. This strategy opens an avenue for practical applications towards on-chip microcavity CP-OLEDs.

Circularly polarized (CP) light, featuring optical rotatory power and rich angle-independent properties, has attracted increasing attention for a variety of potential applications, such as encrypted information storage[1,2], three-dimensional (3D) displays[3–7], remote sensing[8,9], and chiroptical switches[10–13]. In the case of organic light-emitting diode (OLED) based 3D-display technology, incorporation of extra polarizers is required to produce CP light from unpolarized electroluminescence (EL), leading to large power loss and poor contrast ratio[14,15]. Therefore, the active generation of CP light directly from OLEDs, i.e., CP-OLEDs, is more practical, due to their simple device architectures and energy saving without the need for extra polarizers[5,6,16,17]. So far, the dominant efforts for CP-OLEDs focus on the development of chiral emitters as

[1]Beijing Key Laboratory for Optical Materials and Photonic Devices, Department of Chemistry, Capital Normal University, 100048 Beijing, People's Republic of China. [2]Department of Physics and Center for Optoelectronics and Photonics Paderborn (CeOPP), Universität Paderborn, Warburger Strasse 100, 33098 Paderborn, Germany. [3]Institute of Molecule Plus, Tianjin University, and Collaborative Innovation Center of Chemical Science and Engineering (Tianjin), Tianjin 300072, PR China. [4]Wyant College of Optical Sciences, University of Arizona, Tucson, AZ 85721, USA. [5]These authors contributed equally: Jichao Jia, Xue Cao. ✉e-mail: liaoqing@cnu.edu.cn; hbfu@cnu.edu.cn

the active layer for CP EL, including chiral molecules and metal-organic complexes[3,18–20], as well as achiral conjugated polymers with chiral sidechains and chiral dopants[4,21–24]. Nevertheless, high-performance CP-OLEDs remain a great challenge hindered by three stumbling blocks: (1) the CP EL materials are rare, limited by difficult molecular design and synthesis[25]; (2) the key parameter, dissymmetry factor ($g_{EL}$) of the CP-OLEDs, is still relatively low, in the range of $10^{-3}$–$10^{-1}$, except for a few examples which exhibit high $g_{EL}$ up to 1.0 by using chiral transition metal complexes and polymers[4,14,26]; (3) the CP-OLEDs still show lower device performance than conventional OLEDs, for example, the relatively low luminance, inevitably hindering their practical applications[13]. Therefore, the development of CP-OLEDs with high $g_{EL}$, high luminance and of those that are chiral-emitter free is a key issue to be addressed.

The polarization degree of freedom is intrinsic to the nature of light and it can interact with the light field's orbital angular momentum. This is called photonic spin-orbit interaction (SOI) and is reminiscent of SOI in electronic systems with the photon's pseudospin mimicking the electron's spin. Photonic SOI has been widely investigated in inorganic systems, such as graphene, transition-metal dichalcogenide, and metasurface materials[27–29], and brings about abundant applications in optoelectronics ranging from classical information processing to the quantum optical regime[30–32]. The realization of photonic SOI requires the breaking of inversion symmetries in solid-state systems. Contrary to inorganic materials, organic molecular assemblies, especially organic single crystals (OSCs), have highly ordered and anisotropic molecular packing arrangement and therefore anisotropic refractive index and show birefringence. Recently, the tunability of both energy and polarization of the confined photonic modes have been reported in liquid-crystal-filled[33] and OSC-filled[34] birefringent organic cavities. In particular, when two photonic modes with orthogonal linear-polarization and opposite parity are close to resonance, Rashba-Dresselhaus (RD) SOI emerges with the characteristic feature of left- and right-handed CP dispersions[33,34]. Regrettably, the CP-splitting phenomenon due to RD-SOI is demonstrated in most cases in passive reflection mode. We anticipate that combining a birefringent OSC cavity with OLED architecture might lead to the realization of artificial RD SOI, producing CP EL without the need of chiral emitters.

In the present work, we demonstrate a chiral-emitter free CP-OLED with high $g_{EL}$ and high luminance by embedding a thin two-dimensional OSC (2D-OSC) of 1,4-bis((E)−2,4-dimethylstyryl)-2,5-dimethylbenzene (6M-DSB) sandwiched between two silver layers which serve as two mirrors forming a planar microcavity and meanwhile as two electrodes in OLED configuration. Introducing birefringent 6M-DSB 2D-OSC into a planar microcavity modulates the polarization of the confined photonic modes through artificial RD SOI. Thanks to high emission efficiency and high carrier mobility of 6M-DSB OSC[35–41], chiral-emitter free CP-OLEDs are demonstrated with a high $g_{EL}$ of 1.1 and a maximum luminance of about 60,000 cd/m², which are among the best performances of CP-OLEDs. This unique top-emitting microcavity OLED architecture with active CP EL based on RD SOI demonstrates a promising strategy for future display and communication applications.

## Results

### Sample preparation and properties

A schematic of our device is shown in Fig. 1a. The microcavity CP-OLEDs were fabricated with a top-emitting device architecture: silicon wafer/Ag (200 nm)/6M-DSB OSC (990 nm)/CsF (10 nm)/Ag (35 nm) (see details in Supplementary Figs. S1–S3), where the bottom Ag and the top CsF/Ag films played the role of hole and electron injection layers, respectively. Because of the excellent reflective properties of the silver films (that is, the reflectivity of the silver film with the thickness of 200 nm is more than 99% and that of 35 nm is about 50%),

the parallel silver electrodes also double as the high-quality reflectors to form an optical Fabry-Pérot planar microcavity. Thin 2D-OSCs of 6M-DSB were chosen as the optical medium inside the microcavity, because of its giant anisotropy of the refractive index along Y (2.40) and X (1.95) directions (see Fig. 1a and also text below)[42,43]. Furthermore, its semiconducting feature and high solid-state photoluminescence (PL) quantum yield (PLQY = 0.9931, see details in Supplementary Fig. S6) assure efficient electron and hole injection for high-performance EL.

Thin 2D-OSCs of 6M-DSB were prepared by the physical vapor transport method, with the lateral size in a millimeter scale and a thickness of about 990 nm (Supplementary Fig. S7). The uniform and smooth surface of 6M-DSB 2D-OSCs with a roughness less than 1.5 nm pave the way for fabricating microcavity CP-OLEDs. As-prepared 2D-thin-OSCs of 6M-DSB were then transferred on the 200-nm Ag film pre-deposited on silicon wafer. Finally, CsF/Ag films were vacuum deposited sequentially through a mesh-mask, giving rise to an array of microcavity CP-OLEDs on the surface of thin 2D-OSCs. Figure 1a also presents the top-view bright-field photograph of patterned microcavity CP-OLEDs. It can be seen that subunits of CP-OLED array are in $40 \times 40\ \mu m^2$ and separated by 20 μm to avoid cross-talk and short circuit. The thin 2D-OSCs of 6M-DSB adopt a lamellar structure with the crystal (001) plane parallel to the substrate (Fig. 1b and Supplementary Fig. S8)[44]. Within (001) plane, nearly planar 6M-DSB molecules stack in a brickwork arrangement among their short-axis with the nearest π-π distance ca. 3.6 Å. Their molecule long-axis is tilted at an angle of 63° respect to the substrate[44]. It can be seen from Fig. 1b that this brickwork arrangement brings about significant anisotropic molecular packing arrangements (molecular packing density) along and parallel to the π−π interaction (defined as Y- and X-direction, respectively), thus leading to a strong anisotropy of the refractive index. We associate the linear polarizations of the cavity modes along X- and Y-direction as X- and Y-polarization, respectively.

### Principle of SOI and experimental realization

In theory, such a birefringent microcavity can be approximately described by an effective Hamiltonian $H(\mathbf{k}) = H_{TETM} + H_{RD} + H_{XY}$, where $H_{TETM}$ describes the intrinsic transverse-electric-transverse-magnetic (TE-TM) splitting of the cavity modes[45], $H_{RD} = -2\alpha\hat{\sigma}_z k_y$ is the RD Hamiltonian[33,34,46], giving rise to a spin-splitting along $k_y$ direction with the strength $\alpha$, and $H_{XY} = \beta_0\hat{\sigma}_x$ is the Hamiltonian representing the XY splitting[45], i.e., the energy splitting ($\beta_0$ at $\mathbf{k} = 0$) of the perpendicularly linearly polarized modes (X- and Y-polarizations) with opposite parity (here, we define it as $\beta_0 = E_X - E_Y$, where $E_X$ and $E_Y$ are the ground state energies of X and Y modes of opposite parity). The above effective Hamiltonian in the circular polarization basis can be written in the form of a 2 × 2 matrix:

$$H(\mathbf{k}) = \begin{pmatrix} E_0 + \frac{\hbar^2}{2m}\mathbf{k}^2 - 2\alpha k_y & \beta_0 + \beta_1\mathbf{k}^2 e^{2i\varphi} \\ \beta_0 + \beta_1\mathbf{k}^2 e^{-2i\varphi} & E_0 + \frac{\hbar^2}{2m}\mathbf{k}^2 + 2\alpha k_y \end{pmatrix}, \quad (1)$$

where $E_0$ is the energy of the ground state, $m$ is the effective mass of cavity photons, $\beta_1$ is the strength of the TE-TM splitting, and $\varphi$ ($\varphi \in [0, 2\pi]$) is the polar angle.

The left panel of Fig. 1c depicts the energy ($E$) vs. momentum ($k_y$) dispersion in an isotropic microcavity. Here, two orthogonally polarized cavity modes with the same parity ($X_n$, $Y_n$) degenerate at $k_y = 0$, with $\beta_0 > 0$ ($\beta_0 = 0$ means two orthogonally polarized cavity modes with different parity at $k_y = 0$ to be resonant). Once 2D-OSCs of 6M-DSB were inserted, the anisotropy of its refractive index leads to a birefringent microcavity, in which the splitting between X- and Y-polarized modes at $k_y = 0$ occurs, leading to a reduction of $\beta_0$ (see the middle panel of Fig. 1c). By carefully tuning the cavity length, i.e., the thickness of the 2D-OSCs, two orthogonally linearly polarized

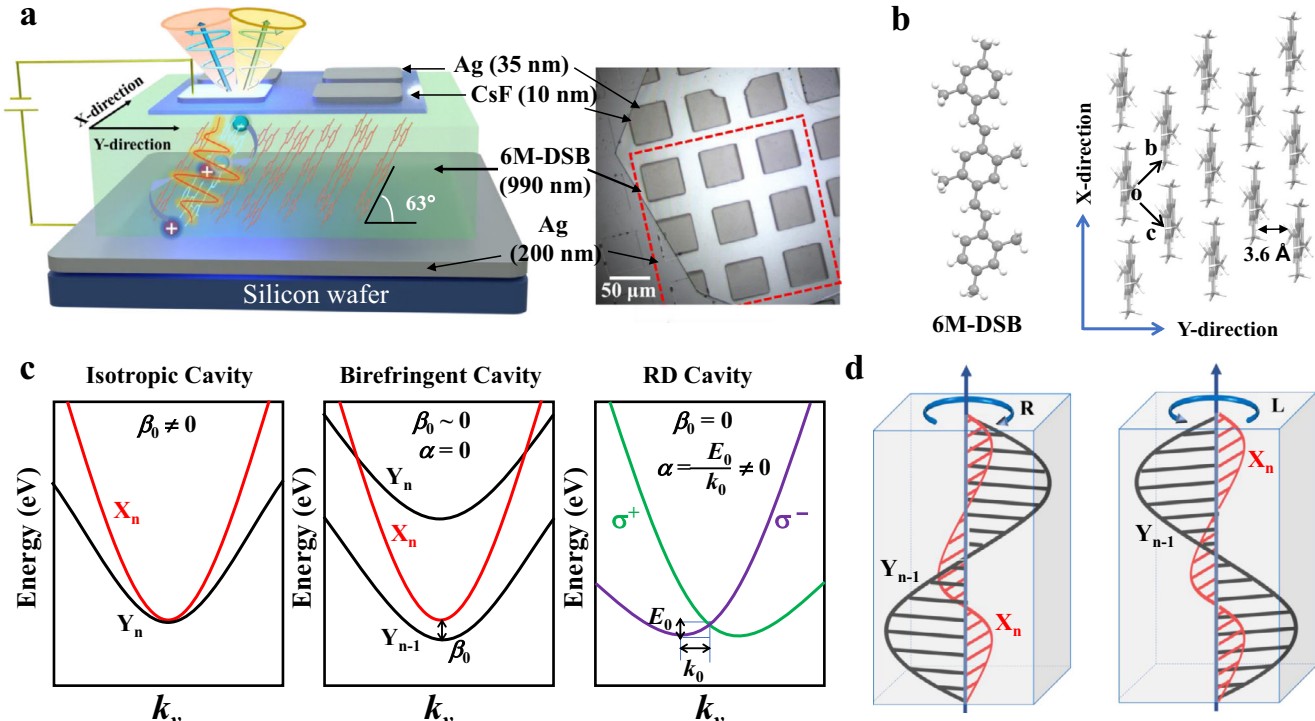

**Fig. 1 | Sample architecture and principle of SOI. a** Schematic diagram of the microcavity CP-OLED device structure. The bottom Ag (200 nm) and the top Ag (35 nm)/CsF (10 nm) films serve as the hole and electron injection layers for the OLED and meanwhile function as two mirrors forming a Fabry-Pérot planar microcavity. The 6M-DSB OSCs serve as the emitting layer of the OLED and as birefringent optical medium necessary for the realization of SOI in the microcavity. Right panel: the top-view bright-field of patterned microcavity CP-OLEDs. **b** Left: Molecular structure of 6M-DSB from single-crystal data, showing the enhanced planarity. Right: brickwork molecular packing arrangement within the (001) crystal plane, viewed perpendicular to the microribbon top-facet. The nearest

intermolecular distance is around 3.6 Å. **c** Left: two orthogonally linearly polarized modes with the same parity in an isotropic microcavity. Middle: the dispersion of two orthogonally linearly polarized modes in an anisotropic microcavity. Right: RD SOI emerges when two orthogonally linearly polarized modes with opposite parity are resonant. **d** The resonant X- and Y-polarized cavity modes of opposite polarity. The 6M-DSB crystal in the microcavity hence acts as a half-wave plate, and the intrinsic mode polarization of the light-emitting side of the mirror turns into a circle, corresponding to left-handed and right-handed circular polarizations, respectively.

cavity modes with opposite parity ($X_n$, $Y_{n-1}$) approach each other at $k_y = 0$, that is, $\beta_0$-0. In the resonant case ($\beta_0 = 0$), a clear spin-splitting appears along $k_y$ direction due to RD SOI with $\alpha \neq 0$ (see right panel of Fig. 1c) as predicted by Eq. (1) and therefore linear polarizations change to circular polarizations. That is, at resonance of $X_n$ and $Y_{n-1}$, the phase difference between the X- and Y-polarized modes across the intra-cavity anisotropic 2D-OSC rotate by π, such that the anisotropic 2D-OSC act as a half-wave plate and result in a change of the eigenmode polarization at the mirror interfaces from linear to circular polarizations (Fig. 1d)[33].

To experimentally demonstrate the above theoretical prediction, we firstly measured the unpolarized angle-resolved reflectivity (ARR) of our microcavity CP-OLED along Y-direction. The reflectivity is plotted as a function of wavelength (or energy) and angle (or momentum $k_y$), measured by using a home-made micro-scale ARR measurement system (Supplementary Fig. S4). Figure 2a presents two sets of modes with distinctive curvatures. The black and red dashed lines are simulated dispersions, in good agreement with experimental results. The simulated refractive indices of the two cavity modes (that is, 1.95 for X-polarized modes and 2.40 for Y-polarized modes) also support the fact of the giant anisotropy of the thin 2D-OSCs of 6M-DSB (Supplementary Fig. S9). We measure the polarized reflection spectrum and obtain accurate refractive index values (Supplementary Fig. S15), which are in line with our simulation results. We also perform angle-resolved PL (ARPL) measurements under the excitation of a 405-nm continuous-wave laser. The ARPL (Fig. 2b) and ARR (Fig. 2a) spectra show exactly the same dispersions. Furthermore, polarization-

resolved ARPL experiments have been carried out by adding a quarter-wave plate and a linear polarizer along the detection optical path (Supplementary Fig. S5), which allows us to distinguish the polarization of the relevant optical modes. The modes with larger (red dashed lines) and smaller (black dashed lines) curvatures are X- and Y-polarization corresponding to X- and Y-direction, respectively, of the 2D-OSCs of 6M-DSB, which are both well consistent with the calculated results of the cavity modes by using the 2D cavity photon dispersion relations[34].

The energy splitting values $\beta_0$ between the X- and Y-polarized modes present wavelength-dependent characteristics, for example, $\beta_0$ is about 23, 0, and −21 meV for near 448, 497, and 558 nm, respectively, as shown in Fig. 2a, b. Different from other successive dispersion branches (such as at 448 and 558 nm), the crossing point appears at 497 nm and $k_y = 0$ originated from $X_9$ and $Y_8$ branches (black dotted circle in Fig. 2a), which suggests that the RD SOI emerges in this device. To investigate the polarization property of these cavity modes, we further measured the Stokes vector components[47] to analyze the pseudospin behaviors. The $S_1$ components of the Stokes vector of $X_8$ and $Y_7$ branches show strongly linearly polarized PL emission (Fig. 2e), while their corresponding $S_3$ components exhibit relatively weak (Fig. 2f). As X- and Y-polarized modes approach, the near resonant $X_9$ and $Y_8$ branches at $k_y = 0$ enhances the RD SOI, leading to the clear splitting of the paraboloids in $k_y$ direction (Fig. 2b). The $S_3$ components present two separate circles with opposite signs and they are highly CP with much higher polarization degree (Fig. 2d), while the $S_1$ components become very weak (Fig. 2c). Therefore, the PL-active CP

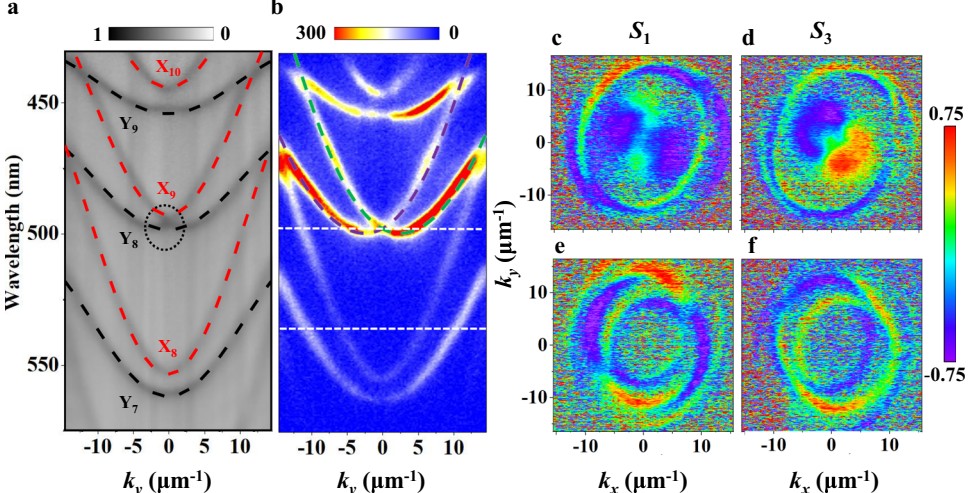

**Fig. 2 | Reflection and PL measurements. a** ARR and **b** ARPL of the microcavity with the organic-layer thickness of 990 nm. The units of the color bars are arbitrary. The dotted curves in **a** represent the simulated X and Y cavity modes, respectively. The dashed curves in **b**, indicating the split CP modes, are calculated by solving Eq. (1) with the parameters $E_0 = 2.495$ eV (497 nm), $m = 2.2 \times 10^{-5} m_e$ ($m_e$ is the free electron mass), $\beta_0 = 0$, $\beta_1 = 0.7$ meV, $\alpha = 37$ eV Å. **c–f** Cross-section maps of the 2D tomography at 497 and 535 nm in momentum space, corresponding to the dashed lines in **b**, respectively. $S_1$ components of the Stokes vector at 497 nm (**c**) and 535 nm (**e**), $S_3$ components of the Stokes vector at 497 nm (**d**) and 535 nm (**f**).

emission due to the RD SOI has been clearly demonstrated in our microcavity OLEDs.

## EL performances of the CP-OLED

The EL performances of our microcavity CP-OLED were then investigated. Figure 3a shows the energy level diagram of the OLED architecture. Ag and Ag/CsF are used as the upper and lower electrodes to inject holes and electrons, respectively. Under a certain voltage, a device achieves uniform and bright EL and edge waveguide (Fig. 3b and Supplementary Fig. S10). We collect the EL emissions from the upper electrode and the crystal edge and compare them with the PL spectrum of 6M-DSB OSCs (Fig. 3c). The waveguided EL light emitted from the OLED is trapped in single crystals in the form of a whisper gallery mode and then propagates out of the crystal edge. So, this EL spectrum (middle panel of Fig. 3c) is coincident with the PL spectrum of the OSCs (upper panel of Fig. 3c). Notably, the EL spectrum from the upper electrode (bottom panel in Fig. 3c) exhibits multiple microcavity resonant peaks, and these peaks' wavelengths are obviously consistent with the PL spectrum of the OSCs. This indicates that the EL emission comes from the OSC active layer of the OLED and is regulated by the microcavity. The EL performances of a single-crystal OLED based on 6M-DSB single crystals are shown in Fig. 3d, e. The maximum luminance and current efficiency of about 60,000 cd/m² and 1.48 cd/A, respectively, were obtained. Thanks to the good crystal quality of the OSCs, our OLED can withstand high voltage of tens of volts (that is, current density of 22.7 A/cm²) and stably emit light without any damage. The highest luminance of our OLED reaches 60,000 cd/m² at the current density of 7.6 A/cm², which is one of the highest luminance of single-crystal OLEDs.

In order to study the influence of the RD SOI on the EL emission, we perform the angle-resolved EL (AREL) measurement for our OLED. The angle range of the AREL spectrum is only ±15° limited by numerical aperture of the microscope lens in our setup. The obtained AREL spectrum also shows a clear RD spin-splitting, which matches well with the EL spectrum (Supplementary Fig. S11). We further performed the CP (σ+ and σ−) AREL spectroscopy. Expectedly, the EL emission exhibit strong left- and right-handed circular polarization in the vicinity of 497 nm at $k_y = 0$ as shown in Fig. 4a, b, respectively, which agrees to the result of the ARPL as shown in Fig. 4c, d. This strongly testifies that the RD SOI occurs at the condition of electrical excitation and induces

the active CP emission due to the spin splitting. Correspondingly, the EL emission spectra of unpolarized, σ+ and σ− are shown in Fig. 4e. The left- and right-handed CP emissions are exactly coincident with the above analysis of the cavity modes.

We calculate the RD spin-splitting parameters ($\alpha = 2E_0/k_0$)[48], as a typical value for the feature of RD SOI, at different photonic modes and show in Fig. 4f (blue triangles). Two modes with different polarizations and opposite parity are brought into resonance and give rise to wavelength-dependent $\alpha$ values of the RD effect (Supplementary Figs. S12–S14). In the vicinity of the resonance ($X_9$ and $Y_8$ modes), we obtain a giant RD parameter of $\alpha = 52$ eV Å, which is much larger than that reported in liquid-crystal optical cavities[33]. The key parameters of circular polarization asymmetry factors ($g_{EL}$ for EL and $g_{lum}$ for PL) are usually employed to characterize the performance of CP-OLEDs and can be defined as $g_{EL}$ (or $g_{lum}$) = $2 \times (I_L - I_R)/(I_L + I_R)$, where $I_L$ and $I_R$ correspond to the intensities of left- and right-handed polarization, respectively[49]. The obtained $g_{EL}$ and $g_{lum}$ on the different photonic modes are presented in Fig. 4f. At the resonance wavelength, $g_{EL}$ and $g_{lum}$ reach the maximum values of about 1.4 and 1.1, respectively, which are both the best performances of CP-OLEDs (Supplementary Table S1).

## Discussion

In summary, we propose a strategy for designing chiral-emitter-free single-crystal CP-OLEDs with high luminance and large $g_{EL}$ value through introducing artificial RD SOI into optical microcavities. Thanks to high emission efficiency and high carrier mobility of the 6M-DSB OSC, our microcavity CP-OLEDs exhibit a maximum luminance of exceeding 60,000 cd/m² and a large $g_{EL}$ value of 1.1, which are among the best performances of single-crystal CP-OLEDs. The introduction of OSC to such EL devices brings about two advantages: (1) the highly ordered molecular stacking arrangement in single crystals can result in anisotropic refractive index in different directions, which benefit the occurrence of the RD effect. (2) The high current density in single-crystal OLEDs (e.g., 69.25 A cm⁻² in our case, which is three orders of magnitude higher than that of thin-film OLEDs) also makes them potential candidates also for the future realization of organic electrically pumped solid-state lasers. Our strategy provides a promising design for efficient single-crystal CP-OLEDs, which may greatly promote the development of CP-OLEDs with use in future 3D display applications.

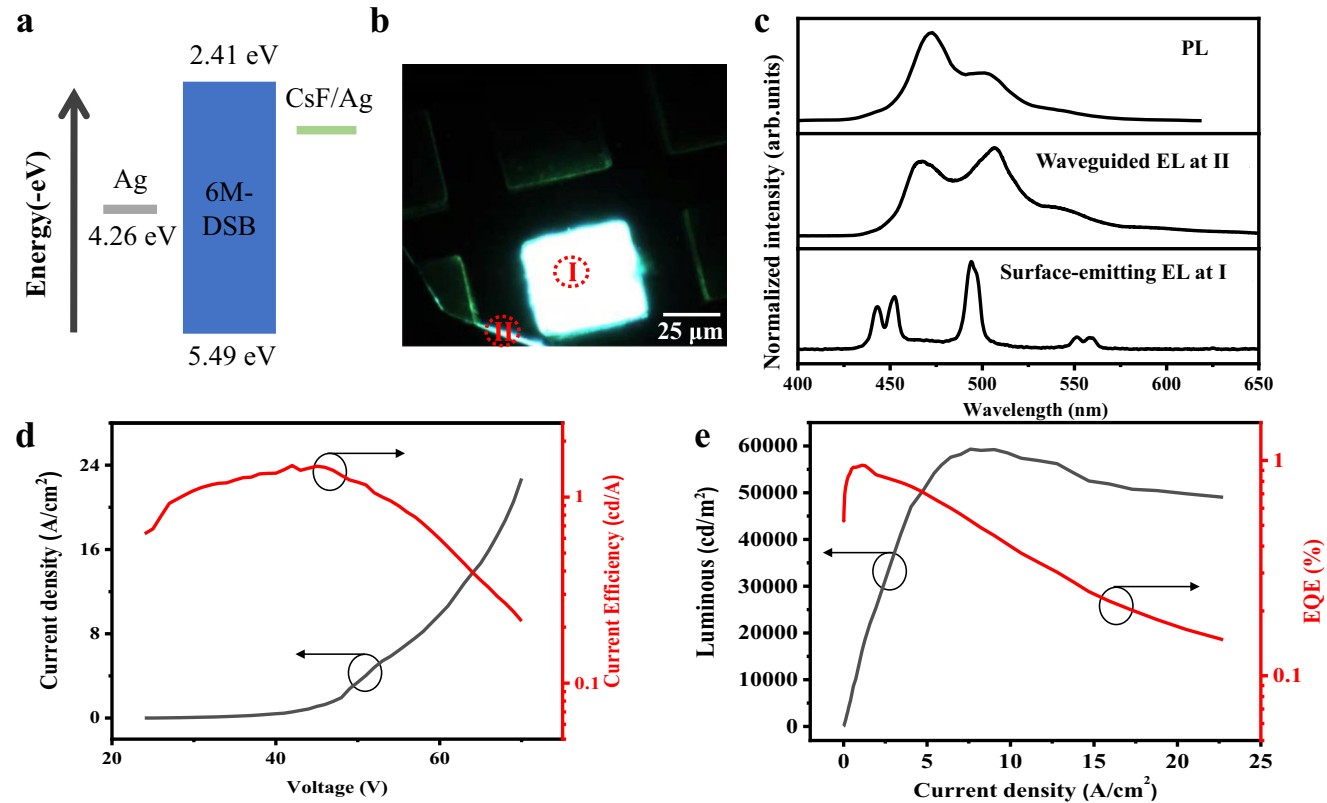

**Fig. 3 | EL architecture and performances. a** Energy level diagram of the device. **b** Photomicrograph of the 6M-DSB single-crystal OLED array under EL. **c** Upper: PL spectrum of the 6M-DSB crystal. Middle: spectrum of the crystal edge waveguide under EL as shown in **b**. Lower: spectrum of light emanating from the microcavity under EL as shown in **b**. **d** Dependence of the current density (black line) and the current efficiency (red line) on the voltage. **e** Dependence of the luminescence (black line) and the EQE (red line) on the current density.

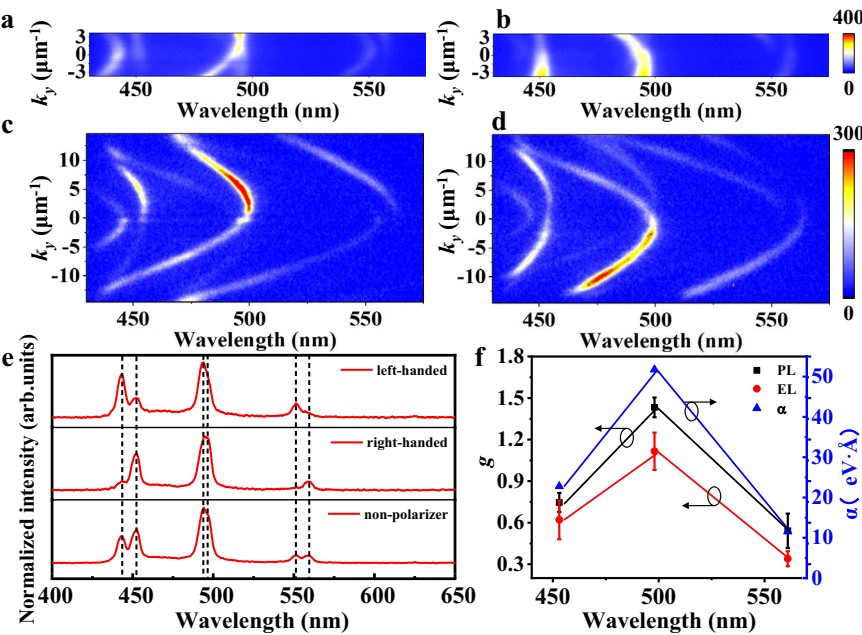

**Fig. 4 | Angle-resolved EL measurements.** Polarized angle-resolved spectra of EL (**a, b**) and PL (**c, d**). The units of the color bars are arbitrary. **e** EL emission spectra of unpolarized, σ+ and σ− components. **f** Wavelength-dependent spin-splitting coefficient $\alpha$ of the RD effect and circularly polarized luminescence dissymmetry factor $g$ of PL and EL with standard deviation error bars.

## Data availability

The data that support the findings of this study are available from the corresponding author upon reasonable request.

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

## Acknowledgements

This work was supported by the National Key R&D Program of China (Grant No. 2018YFA0704805 (Q.L.), 2018YFA0704802 (Q.L.) and 2017YFA0204503 (H.F.)), the National Natural Science Foundation of China (22150005 (Q.L.), 22090022 (H.F.), 21833005 (H.F.) and 21873065 (Q.L.)), the Natural Science Foundation of Beijing, China (KZ202110028043 (H.F.)), Beijing Talents Project (2019A23 (Q.L.)), Capacity Building for Sci-Tech Innovation-Fundamental Scientific Research Funds (H.F.), Beijing Advanced Innovation Center for Imaging Theory and Technology (H.F.). The authors thank Dr. H.W. Yin from Ideaoptics Inc. for the support on the angle-resolved spectroscopy measurements.

## Author contributions

J.J., X.C., J.D. and Q.L. designed the experiments and performed experimental measurements. X.M. and S.S. performed the theoretical calculation and analysis. X.M., Q.L. and H.F. wrote the manuscript with contributions from all authors. Q.L., J.Y. and H.F. supervised the project. All authors analyzed the data and discussed the results.

## Competing interests

The authors declare no competing interests.
