## [Peer review file · Nature Communications]

REVIEWER COMMENTS

Reviewer #1 (Remarks to the Author):

The work by Jia et al. presents circularly polarized EL emission from OLEDs that do not rely on chiral emitters but are based on a microcavity structure made of an organic single crystal sandwiched between silver layers. Given the difficulty of developing efficient chiral emitters, the proposed approach is interesting and could eventually merit publication in Nature Communication provided that authors properly address the following issues:

1. Figure 1(a) would rather be revised to better show the structure of 6M-DSB and its molecular arrangement within the crystal. Definition of crystalline planes may also be specified if relevant. In addition, information provided by Fig. 1(b) appears not so critical and would rather be replaced with an energy diagram, etc. Alternatively, the bright-field image may be combined with Fig. 1(a), and the energy diagram may be added as Fig. 1(b).
2. Please discuss on whether the thickness of 6M-DSB crystal influences the dissymmetry factor, etc. or not. If so, please comment on how to control its thickness.
3. Please discuss on what would happen to the overall CP performance if amorphous organic layers are added for carrier injection, exciton blocking, and/or carrier transport layers.
4. Simply from the OLED perspectives, the present device geometry appears to go against its development path because the initial organic EL was discovered with anthracene crystal. The major breakthrough in the early OLED history was to replace the crystal with thin films and introduce multilayer structures for balanced carrier injection and confinement of emitting layer. In this respect, the present device geometry is too far from ideal in terms of device efficiency and operation voltage. Authors are welcomed to add a discussion on this issue and provide a strategy that can make the present approach a practically viable one.

Reviewer #2 (Remarks to the Author):

In this manuscript, Jia et al. reported a circularly polarized OLED based on photonic spin-orbit interactions by embedding an organic single crystal (OSC) between two silver layers forming a microcavity. OSCs possess a highly ordered molecular arrangement with an anisotropic refractive index of birefringence, which can provide a birefringent OSC cavity to realize artificial RD SOI and produce CP EL without using chiral emitters. However, there are still some scientific ambiguities that haven't been clarified clearly. In my thought, the current status of this work is not suitable for publication in Nature Communication. Please consider them seriously as follows:

1. The authors should provide more details of the calculation for the refractive indexes along X and Y directions shown in Figure S2. And more experiments like polarized reflectance spectra are suggested to be measured for estimating the refractive indexes with accurate values.
2. There are missing data: the authors provide the solid-state PL quantum yield of 0.82, but there is no supporting data for this value.
3. The 6M-DSB OSC adopts a lamellar structure with the crystal (001) plane parallel to the substrate. Without XRD or SEAD data, how did the author determine the molecular arrangement in the crystal? Also, the brickwork arrangement of 6M-DSB shown in fig. 1a is not clear enough for observation which is suggested to be modified.
4. As we know, OSCs possess an anisotropic refractive index of birefringence due to their highly ordered molecular arrangement. Why did the authors choose 6M-DSB OSC in this work? The authors have reported perylene crystals with the same ability to control the spin-orbit interaction. Do all the OSCs have the same ability to realize the exciton-mediated optical spin-orbit interaction?
5. The authors demonstrate that two orthogonally linearly polarized cavity modes with opposite parity can be realized by carefully tuning the cavity length, i.e, the crystal thickness. How can they precisely control the crystal thickness? What is the relation between the cavity length and the RD SOI? Whether the RD SOI will disappear while the cavity length decreases or increases.
6. According to the Rashba-Dresselhaus SOI theory, the Hamiltonian contains three items. When the HTETM can dominate splitting behavior and $\beta_1 = 0$? Does it depend on the cavity length or the refractive index of birefringence?
7. Why the obtained waveguided EL and surface-emitting EL are different? As the optical waveguide of OSC, the two EL should be similar to each other and have the same circularly polarized behavior. The authors should provide more experimental details to identify.
8. According to the ARPL, there is a giant RD spin-splitting in the vicinity of the resonance of X9 and Y8 modes (497 nm). However, there is no exchange between the modes near 448 nm and 558 nm, indicating that the RD effect is not dominating. Why do the EL peaks at 448 nm and 558 nm show the same spin-splitting as shown in the left- and right-handed EL spectra? Are they TE and TM polarization?
9. It cannot be confirmed whether the observed EL spectra are circularly polarized or linearly polarized as TE or TM polarization. The author did not describe the measuring process. Are a half-wave plate and a quarter-wave plate together placed in front of spectrometer to obtain the circular polarization? The authors should provide more experiments to prove the obtained EL is circular polarization.

10. Also, single-crystal OLEDs show a high turn-on voltage of about 60 V, which is not suitable for practical application. It is suggested to add some modified layered to improve the OLED EL performance.

11. The author should mind some typos in the manuscript, such as "14000 cd/cm²"...

Reviewer(s)' Comments to Author:

Reviewer #1

Comments:

General Comment: *The work by Jia et al. presents circularly polarized EL emission from OLEDs that do not rely on chiral emitters but are based on a microcavity structure made of an organic single crystal sandwiched between silver layers. Given the difficulty of developing efficient chiral emitters, the proposed approach is interesting and could eventually merit publication in Nature Communication provided that authors properly address the following issues:*

Response: We thank the reviewer for the very positive appreciation of our work. We are very thankful to the reviewer for his/her careful reading and providing valuable suggestions which helped us to improve our manuscript significantly.

Comment 1: *Figure 1(a) would rather be revised to better show the structure of 6M-DSB and its molecular arrangement within the crystal. Definition of crystalline planes may also be specified if relevant. In addition, information provided by Fig. 1(b) appears not so critical and would rather be replaced with an energy diagram, etc. Alternatively, the bright-field image may be combined with Fig. 1(a), and the energy diagram may be added as Fig. 1(b).*

Response: We thank the reviewer for this valuable suggestion and have revised Figure 1 (as also shown in Figure R1 below). The bright-field image has been combined with Figure 1a. Figure 1b shows the structure of 6M-DSB and its molecular arrangement. Brickwork molecular packing arrangement within the (001) crystal plane, viewed perpendicular to the microribbon top-facet. The nearest intermolecular distance is $d \approx 3.6 \text{ \AA}$. However, the energy diagram is not added into Figure 1b to avoid repeating the same information because it has been shown in Figure 3a.

Figure R1 (Figure 1). (a) Schematic diagram of the microcavity CP-OLED device structure. The bottom Ag (200 nm) and the top Ag (35 nm)/CsF (10 nm) films serve as the hole and electron injection layers for the OLED and meanwhile function as two mirrors forming a Fabry-Pérot planar microcavity. The 6M-DSB OSCs serve as the emitting layer of the OLED and as birefringent optical medium necessary for the realization of SOI in the microcavity. Right panel: the top-view bright-field of patterned microcavity CP-OLEDs. (b) Left: Molecular structure of 6M-DSB from single-crystal data, showing the enhanced planarity. Right: brickwork molecular packing arrangement within the (001) crystal plane, viewed perpendicular to the microribbon top-facet. The nearest intermolecular distance is $d \approx 3.6 \text{ \AA}$. (c) Left: two orthogonally linearly polarized modes with the same parity in an isotropic microcavity. Middle: the dispersion of two orthogonally linearly polarized modes in an anisotropic microcavity. Right: RD SOI emerges when two orthogonally linearly polarized modes with opposite parity are resonant. (d) The resonant X- and Y-polarized cavity modes of opposite parity. The 6M-DSB crystal in the microcavity hence acts as a half-wave plate, and the intrinsic mode polarization of the light emitting side of the mirror turns into a circle, corresponding to left-handed and right-handed circular polarizations, respectively.

Comment 2: Please discuss on whether the thickness of 6M-DSB crystal influences the dissymmetry factor; etc. or not. If so, please comment on how to control its thickness.

Response: The thickness of 6M-DSB crystal can be controlled by the deposition temperature of physical vapor deposition in our experiments. The sublimation temperature of 6M-DSB is 230 °C and the flow rate of the inert gas is set to 100 sccm·min⁻¹. For example, the crystal thickness obtained is about 380 nm when the

deposition temperature is set to 150 °C. As the deposition temperature decreases to 110 °C, the crystals with a thickness of about 835 nm are obtained. When deposition temperature reaches 90 °C, the crystal thickness is about 1325 nm. The plot of the crystal thickness versus the deposition temperature is shown in Figure R2 below. We have added the results in the revised Supplementary Materials (Scheme S3, Page 4).

Figure R2 (Scheme S3). Dependence of the crystal thickness on the deposition temperature.

In order to investigate the relation between the thickness of 6M-DSB crystal and the dissymmetry factor for circularly polarized emission, we chose three typical 6M-DSB crystals with the thickness of 835 nm, 990 nm and 1325 nm and performed their angle-resolved photoluminescence and electroluminescence spectra. As shown in Figure R3 (also see Figure S7, Page 15), the location of the crossing point, characteristic feature of the RD effect, shifts from shorter wavelength to longer wavelength with increasing crystal thickness. We calculated the dissymmetry factors of these three devices and the results are summarized in Table S1. The photoinduced and electroinduced dissymmetry factors are almost the same as those of 990-nm devices in the manuscript. Therefore, it can be concluded that the crystal thickness affects only the position of the crossing point induced by the RD spin-splitting not the dissymmetry factor. We have added the results in the revised Supplementary Materials (Figure S7 and Table S1, Pages 15-16).

Figure R3 (Figure S7). Angle-resolved photoluminescence (a,g,m) and electroluminescence spectra (d,j,p) of the organic-crystal microcavity with the crystal thickness of 835 nm, 990 nm and 1325 nm, respectively. The white dashed rectangles mark the positions where the RD effect occurs. The corresponding circularly polarized angle-resolved spectra of PL (b,c) and EL (e,f) with the organic-crystal thickness of 835 nm, PL (h,i) and EL (k,l) with the organic-crystal thickness of 990 nm, and PL (n,o) and EL (q,r) with the organic-crystal thickness of 1325 nm.

Table R1 (Table S1). Summary of PL and EL Performance of the Devices with Different Thicknesses.

Thickness (nm)	V _{on} (V)	Current density (A/cm ²)	Luminous (cd/m ²)	EQE (%)	g _{EL}	g _{PL}
835	19	36.77	60829	0.84	-1.20 ≤ g _{EL} ≤ 1.14	-1.27 ≤ g _{PL} ≤ 1.42
990	20	22.7	59324	0.96	-1.25 ≤ g _{EL} ≤ 1.11	-1.30 ≤ g _{PL} ≤ 1.43
1325	31	10.01	53868	1.008	-1.23 ≤ g _{EL} ≤ 1.07	-1.31 ≤ g _{PL} ≤ 1.40

Comment 3: Please discuss on what would happen to the overall CP performance if amorphous organic layers are added for carrier injection, exciton blocking, and/or carrier transport layers.

Response: We thank the reviewer for the suggestion. We performed the CP-OLEDs with some amorphous organic layers added, such as MoO₃ and TPBi (Figure R4). Unfortunately, the obtained device performances are significantly reduced compared to those of our original devices. The detailed performances are summarized in Table R2 and R3. The possible reason might be the mismatch of work functions of electrodes, semiconductors and amorphous organic layers in these devices. The further study to optimize the device configuration and improve the device performance is currently ongoing.

Figure R4. Schematic diagram of the device structure of the microcavity CP-OLEDs.

Table R2. Summary of EL properties of devices with different MoO₃ thicknesses.

Crystal thickness (nm)	V _{on} (V)	Current density (A/cm ²)	Luminous (cd/m ²)	EQE (%)
592	18	69.25	24488	0.26
928	30	28.95	38993	0.20
1294	50	15.56	15972	0.34

Table R3. Summary of EL properties of devices with different TPBi thicknesses.

Crystal thickness (nm)	V _{on} (V)	Current density (A/cm ²)	Luminous (cd/m ²)	EQE (%)
180	15	132.6	15466	0.20
446	13	124.8	23473	0.06
858	31	80.35	17279	0.11

Comment 4: Simply from the OLED perspectives, the present device geometry appears to go against its development path because the initial organic EL was discovered with anthracene crystal. The major breakthrough in the early OLED history was to replace the crystal with thin films and introduce multilayer structures for balanced carrier injection and confinement of emitting layer. In this respect, the present device geometry is too far from ideal in terms of device efficiency and operation voltage. Authors are welcomed to add a discussion on this issue and provide a strategy that can make the present approach a practically viable one.

Response: We thank the reviewer for the suggestion. For the development of single-crystal OLEDs, we are mainly based on the following considerations:

Firstly, organic single crystals have regular morphological structure and highly ordered molecular stacking arrangement. The smooth top/bottom surfaces of organic single crystals are conducive to the formation of optical cavities by combining the two-layer silver mirrors. More importantly, the highly ordered molecular stacking arrangement can result in anisotropic refractive index in different directions of the crystals. This is a necessary condition for the occurrence of RD effect. Therefore, we adopt organic single crystals as the active layer of OLEDs to achieve the CP-OLEDs induced by RD spin-splitting.

Secondly, although organic OLEDs have made great progress in recent years, organic electrically-pumped solid-state laser (OEPSSL), as another important class of organic light sources, still remains a critical challenge. One of the key reasons is the limitation of achieving high current density. In the traditional OLEDs with

multilayer-film structures, the typical operating current density is usually 10 to 100 mA/cm², which is far from ideal in terms of threshold current density (kA cm⁻²) for OEPSSL. We found that organic single crystals are a potential candidate for OEPSSL due to their excellent thermal stability, high carrier mobility, high PLQY and high lasing net gain. In fact, the current density can reach 69.25 A cm⁻² in our 6M-DSB single-crystal OLEDs, which is three orders of magnitude higher than that of thin-film OLEDs. This is another reason why we develop single-crystal OLEDs.

As the reviewer mentioned, the mainstream device geometry of OLEDs is the introduction of multilayer structures for balanced carrier injection and confinement of emitting layer to improve device efficiency and operation voltage. We also fully agree with that. We are trying to improve the device performance through inducing the multilayer film, which is of course beyond the scope of our current work. In the revised manuscript, however, we have added a brief discussion in the Conclusion part (Page 15).

Reviewer #2

Comments:

General Comment: *In this manuscript, Jia et al. reported a circularly polarized OLED based on photonic spin-orbit interactions by embedding an organic single crystal (OSC) between two silver layers forming a microcavity. OSCs possess a highly ordered molecular arrangement with an anisotropic refractive index of birefringence, which can provide a birefringent OSC cavity to realize artificial RD SOI and produce CP EL without using chiral emitters. However, there are still some scientific ambiguities that haven't been clarified clearly. In my thought, the current status of this work is not suitable for publication in Nature Communication. Please consider them seriously as follows:*

Response: We are thankful to the reviewer for his/her careful reading and providing valuable suggestions which helped us to improve our manuscript significantly.

Comment 1: *The authors should provide more details of the calculation for the refractive indexes along X and Y directions shown in Figure S2. And more experiments like polarized reflectance spectra are suggested to be measured for estimating the refractive indexes with accurate values.*

Response: We thank the reviewer for the suggestion. As shown in the reflectivity as a function of wavelength and in-plane wave vector (Fig. 2a), two sets of modes with distinctive curvatures can be seen, which indicates that there are two cavity modes with different effective refractive indices. In order to obtain their effective refractive indices, we have calculated X and Y modes by using the two-dimensional cavity photon dispersion relations (ref. J. Ren, et al., *Laser Photon. Rev.* 2022, 16, 2100252). According to the equation:

$$E_{CMn}(\theta) = \sqrt{\left(E_c^2 \times \left(1 - \frac{\sin^2\theta}{n_{\text{eff}}^2}\right)^{-1}\right) - (n-1) \times l}$$

Where θ represents the incidence angle, $E_{CMn}(\theta)$ is the cavity photon energy of the n^{th} cavity mode as a function of θ , E_c represents the cavity modes energy at $\theta = 0^\circ$, $E_{CM1}(\theta)$ represents the energy of the first cavity modes when $n = 1$, $(n-1) \times l$

represents the energy difference from the first cavity modes. The refractive indices are calculated to be 1.95 and 2.40 for the red and black curves, respectively. The corresponding description has been added in the revised Supplementary Materials (caption of Figure S4, Page 12).

These days, we experimentally measured the refractive indices of the 6M-DSB microbelts following the procedure discussed in *ref.* (Hashimoto S., *et al.*, *Physica Status Solid (b)* 1991, **165**, 277-286). The detailed experimental procedure is described below and has been added into the revised Supplementary Materials (Figure S8-10). Figure R5 (Figure S8) shows the reflection spectrum of the sample with the thickness of 2126 nm, where the red line and black line represent the reflection spectra parallel (Y-) to and perpendicular (X-) to the long direction of the crystal, respectively. The interference peak spacings in both X- and Y-direction does not change significantly.

Figure R5b (Figure S8b, Page 18 of the revised Supplementary Materials) shows X-polarized reflection spectra of X-polarized organic cavities with thickness of 1066 nm, 1488 nm, 1769 nm and 1967 nm, respectively. The interference conditions are given by $2n(\lambda)d = m\lambda$, where $n(\lambda)$ is the refractive index at wavelength λ , m is the order of interference, and d is the crystal thickness. In the reflection spectra, the interference minimum occurs when m is an integer and the maximum occurs when m is a half integer.

The calculated $n(\lambda)$ is shown in Figure R6 (Figure S10, Page 19 of the revised Supplementary Materials). The $n(\lambda)$ in the X-direction rises from 2.31 at 560 nm to 2.52 at 425 nm, and it rises from 1.80 at 560 nm to 1.98 at 425 nm in Y-direction. This is in line with our simulated results.

Figure R5 (Figure S8). (a) Reflection spectra of the 2126-nm-microbelt cavity at X-polarization (black line) and Y-polarization (red line) (b) X-polarized reflection spectra of organic cavities with thickness of 1066 nm, 1488 nm, 1769 nm and 1967 nm.

Figure R6 (Figure S10). The calculated refractive index of 6M-DSB microbelts in X-direction (black dot) and Y-direction (red dot).

Comment 2: There are missing data: the authors provide the solid-state PL quantum yield of 0.82, but there is no supporting data for this value.

Response: We thank the reviewer for the comment. PLQY of 6M-DSB single crystals was measured through an absolute method by using an integration sphere in FLS-1000. As shown in Figure R7, PLQY of 6M-DSB crystals is determined to be 0.9931 at the excitation wavelength of 330 nm. The corresponding description and the figure have been added into the revised manuscript (Lines 3-4, Page 6) and Supplementary Materials (Figure S1, Page 8).

Figure R7 (Figure S1). The experimental PLQY of 6M-DSB crystals.

***Comment 3:** The 6M-DSB OSC adopts a lamellar structure with the crystal (001) plane parallel to the substrate. Without XRD or SEAD data, how did the author determine the molecular arrangement in the crystal? Also, the brickwork arrangement of 6M-DSB shown in fig. 1a is not clear enough for observation which is suggested to be modified.*

Response: We thank the reviewer for the suggestion. We have added the X-ray diffraction (XRD) and selected area electron diffraction (SEAD) results of 6M-DSB single crystals in Fig. S3 (Pages 10-11 of the revised Supplementary Materials) and also showed in Figure R8 below. The XRD pattern was measured by a D/max 2400 X-ray diffractometer with Cu K α radiation ($\lambda = 1.54050 \text{ \AA}$) operated in the 2θ range from 4° to 30° . According to the single-crystal data, the monoclinic crystal of 6M-DSB has the lattice parameters of $a = 4.7533(10) \text{ \AA}$, $b = 5.9928(12) \text{ \AA}$, $c = 18.235(4) \text{ \AA}$, $\alpha = 96.08(3)^\circ$, $\beta = 96.46(3)^\circ$, and $\gamma = 90.15(3)^\circ$. The XRD spectrum of microribbons shows a series of peaks corresponding to the (001) crystal plane with an interplanar spacing of 18.15 \AA (Figure R8a). The observation of high-order diffraction peaks, such as (002)-(005), suggests that the crystal adopts a lamellar structure with the crystal (001) plane being parallel to the substrate. Figure R8c presents SEAD pattern recorded by directing the electron beam perpendicular to the flat surface of a single microribbon. The clearly observed SAED spots and its rectangular symmetry

suggest that 6M-DSB microribbons are single crystals. The squared and triangled sets of SAED spots correspond to (020) and (100) crystal planes with d -spacing values of 6.10 Å and 4.76 Å, respectively, and the blue circled set of SAED spots are attributed to the (110) crystal plane with a d -spacing value of 3.67 Å. Combining the XRD, SAED and TEM results (Figure R8b) together, the 6M-DSB ribbons grow along the [110] crystal direction, bound by (001) and (0-10) crystal planes on the top and bottom surfaces and (1-10) and (-110) crystal planes on the lateral surfaces.

Figure R8 (Figure S3). (a) XRD pattern of ensemble ribbons filtered on the surface of an alumina membrane. (b) TEM image of a typical 6M-DSB microribbon. (c) SAED pattern of the microribbon in (b).

Comment 4: *As we know, OSCs possess an anisotropic refractive index of birefringence due to their highly ordered molecular arrangement. Why did the authors choose 6M-DSB OSC in this work? The authors have reported perylene crystals with the same ability to control the spin-orbit interaction. Do all the OSCs have the same ability to realize the exciton-mediated optical spin-orbit interaction?*

Response: For the first question, we find that 6M-DSB OSCs have not only high PLQY but also excellent electroluminescent (EL) behavior. The good and balanced carrier transport property and high solid-state PL emission endow 6M-DSB OSCs with EL characteristics. However, the perylene crystals we previously reported only have RD effect and no EL properties. Therefore, we choose 6M-DSB OSCs to study their CP-OLEDs in this work.

For the second question, we now find that the anisotropy of the OSCs provides a necessary condition for optical spin-orbit interaction. Nevertheless, the occurrence of

the RD SOI also satisfies this important condition that two photonic modes with orthogonal linear-polarization and opposite parity are close to resonance. Therefore, not all anisotropic crystals can produce the RD effect.

Comment 5: *The authors demonstrate that two orthogonally linearly polarized cavity modes with opposite parity can be realized by carefully tuning the cavity length, i.e., the crystal thickness. How can they precisely control the crystal thickness? What is the relation between the cavity length and the RD SOI? Whether the RD SOI will disappear while the cavity length decreases or increases.*

Response: The thickness of 6M-DSB crystal can be controlled by the method of physical vapor deposition in our experiments. The sublimation temperature of 6M-DSB is 230 °C and the flow rate of the inert gas is set to 100 sccm·min⁻¹. For example, the crystal thickness obtained is about 380 nm when the deposition temperature is set to 150 °C. As the deposition temperature decreases to 110 °C, the crystals with a thickness of about 835 nm are obtained. When deposition temperature reaches 90 °C, the crystal thickness is about 1325 nm. The plot of the crystal thickness versus the deposition temperature is shown in Figure R9. We have added the results in the revised Supplementary Materials (Scheme S3, Page 4).

Figure R9 (Scheme S3). Dependence of the crystal thickness on the deposition temperature.

In order to investigate the relation between the thickness of 6M-DSB crystal and

RD SOI, we chose three typical 6M-DSB crystals with the thickness of 835 nm, 990 nm and 1325 nm and performed their angle-resolved photoluminescence and electroluminescence spectra. According to Figure R10 (also see Figure S7) and Table R4 (Table S1) of the revised Supplementary Materials, it can be concluded that the point of RD SOI will be beyond our detection region of the spectrometer while the cavity length decreases or increases.

Figure R10 (Figure S7). Angle-resolved photoluminescence (a,g,m) and electroluminescence spectra (d,j,p) of the organic-crystal microcavity with the crystal

thickness of 835 nm, 990 nm and 1325 nm, respectively. The white dashed rectangles mark the positions where the RD effect occurs. The corresponding circularly polarized angle-resolved spectra of PL (b,c) and EL (e,f) with the organic-crystal thickness of 835 nm, PL (h,i) and EL (k,l) with the organic-crystal thickness of 990 nm, and PL (n,o) and EL (q,r) with the organic-crystal thickness of 1325 nm.

Table R4 (Table S1). Summary of PL and EL Performance of the Devices with Different Thicknesses.

Thickness (nm)	V _{on} (V)	Current density (A/cm ²)	Luminous (cd/m ²)	EQE (%)	g _{EL}	g _{PL}
835	19	36.77	60829	0.84	-1.20 ≤ g _{EL} ≤ 1.14	-1.27 ≤ g _{PL} ≤ 1.42
990	20	22.7	59324	0.96	-1.25 ≤ g _{EL} ≤ 1.11	-1.30 ≤ g _{PL} ≤ 1.43
1325	31	10.01	53868	1.008	-1.23 ≤ g _{EL} ≤ 1.07	-1.31 ≤ g _{PL} ≤ 1.40

Comment 6: According to the Rashba-Dresselhaus SOI theory, the Hamiltonian contains three items. When the HTETM can dominate splitting behavior and $\beta l = 0$? Does it depend on the cavity length or the refractive index of birefringence?

Response: According to our experience and other's work, the anisotropy, related to the refractive index of birefringence, is a very important factor. However, the thickness of the cavity is also important for example in our case, because it determines whether the two cavity modes with opposite parity and polarization can be resonant. Generally, the RD spin-splitting happens close to, at least not that far away from, $k = 0$, while the TE-TM spin-splitting happens mainly at larger angles. From this point of view one can claim that which spin-splitting dominates in the same system mainly depends on where or which wavelength and angle one measures. In our previous study (J. Ren, *et al.*, *Laser Photon. Rev.* 2022, 16, 2100252), this has been elaborately discussed.

Comment 7: Why the obtained waveguided EL and surface-emitting EL are different? As the optical waveguide of OSC, the two EL should be similar to each other and have the same circularly polarized behavior. The authors should provide more experimental details to identify.

Response: We thank the reviewer for this question. For surface-emitting EL, the electroluminescence couples with the optical cavity and is emitted perpendicularly from the top surface of the OLED. Thus, the surface-emitting EL spectrum displays exactly the same position as the cavity modes. Especially, circularly polarized emission can be obtained from the cavity modes where the RD SOI occurs. In contrast, the waveguided EL is directly emit from the EL of organic crystals without the regulation of the optical cavity, which is because of the limited area of the upper electrode and horizontally-transport EL. Therefore, the EL light without coupling with the optical cavity emits from the edges of the crystal and is consistent with the emission spectrum of 6M-DSB crystal.

***Comment 8:** According to the ARPL, there is a giant RD spin-splitting in the vicinity of the resonance of X9 and Y8 modes (497 nm). However, there is no exchange between the modes near 448 nm and 558 nm, indicating that the RD effect is not dominating. Why do the EL peaks at 448 nm and 558 nm show the same spin-splitting as shown in the left-handed and right-handed EL spectra? Are they TE and TM polarization?*

Response: We thanks the reviewer for the comment. The RD SOI reaches the maximum when two photonic modes with orthogonal linear polarization and opposite parity are close to resonance. When these two photonic modes are slightly separated from each other, the RD spin-splitting does not disappear completely. In our experiments, although the EL peaks at 448 nm and 558 nm show the spin-splitting, the S_3 components of the Stokes vector of these two modes are very weak while the S_1 components (corresponding to linear polarization) become very strong, which indicate that EL emissions from 448 nm and 558 nm are mainly linearly polarized.

***Comment 9:** It cannot be confirmed whether the observed EL spectra are circularly polarized or linearly polarized as TE or TM polarization. The author did not describe the measuring process. Are a half-wave plate and a quarter-wave plate together placed in front of spectrometer to obtain the circular polarization? The authors*

should provide more experiments to prove the obtained EL is circular polarization.

Response: We thank the reviewer for the comment. As shown in Scheme S5 in the revised Supplementary Materials, a quarter-wave plate and a linearly polarized plate are together placed in front of spectrometer to obtain the circular polarization (Figure R11). For circularly polarized EL, the circularly polarized light converts to linearly polarized light through a quarter-wave plate and is detected through a linearly polarized plate. For linearly polarized EL, it can be detected directly through a linearly polarized plate.

Figure R11 (Scheme S5). Experimental setup for the polarization-resolved EL spectra. BS: beam splitter; L1-L4: lenses; M1: mirror. The red beam traces the optical path of the reflected light from the sample at a given angle.

Comment 10: Also, single-crystal OLEDs show a high turn-on voltage of about 60 V, which is not suitable for practical application. It is suggested to add some modified layered to improve the OLED EL performance.

Response: We thank the reviewer for the suggestion. We optimized the growth conditions of the crystals to obtain a large crystal with a very flat surface and a millimeter size. The performance of the device has been improved through the improvement of the crystal quality and the proficiency of the preparation process. The switching on voltage of 20 V and EQE of 0.96% are obtained, and the device has a high brightness of about 60000 cd/m² at the current density of 7.6 A/cm².

In addition, we have added MoO₃ and TPBi as the transport layer, as shown in Figure R12, unfortunately, the obtained device performances are significantly reduced compared to those of our original devices. The detailed performances are summarized in Table R5 and R6. The possible reason might be the mismatch of energy levels in these devices. The further study to optimized the device configuration and improve the device performance is currently going on.

Figure R12. Schematic diagram of the device structure of the microcavity CP-OLEDs.

Table R5. Summary of EL properties of devices with different MoO₃ thicknesses.

Crystal thickness (nm)	V _{on} (V)	Current density (A/cm ²)	Luminous (cd/m ²)	EQE (%)
592	18	69.25	24488	0.26
928	30	28.95	38993	0.20
1294	50	15.56	15972	0.34

Table R6. Summary of EL properties of devices with different TPBi thicknesses.

Crystal thickness (nm)	V _{on} (V)	Current density (A/cm ²)	Luminous (cd/m ²)	EQE (%)
180	15	132.6	15466	0.20
446	13	124.8	23473	0.06
858	31	80.35	17279	0.11

Comment 11: The author should mind some typos in the manuscript, such as “14000 cd/cm2”...

Response: We thank the reviewer for the suggestion. We have carefully revised them in the revised manuscript, such as “cd/m²”.

REVIEWERS' COMMENTS

Reviewer #1 (Remarks to the Author):

Authors have reasonably addressed this reviewer's comments. I believe it is now acceptable for publication in Nature Communications.

Reviewer #2 (Remarks to the Author):

The authors have addressed all the comments by all referees in a satisfactory way. The manuscript is significantly improved. I now recommend its publication with no further questions.

Reviewer(s)' Comments to Author:

Reviewer #1 (Remarks to the Author):

Authors have reasonably addressed this reviewer's comments. I believe it is now acceptable for publication in Nature Communications.

Our response: We thank the reviewer very much!

Reviewer #2 (Remarks to the Author):

The authors have addressed all the comments by all referees in a satisfactory way. The manuscript is significantly improved. I now recommend its publication with no further questions.

Our response: We thank the reviewer very much!